# Activation of the Mechanosensitive Ion Channels Piezo1 and TRPV4 in Primary Human Healthy and Osteoarthritic Chondrocytes Exhibits Ion Channel Crosstalk and Modulates Gene Expression

**DOI:** 10.3390/ijms24097868

**Published:** 2023-04-26

**Authors:** Bibiane Steinecker-Frohnwieser, Birgit Lohberger, Stefan Toegel, Reinhard Windhager, Veronika Glanz, Cornelia Kratschmann, Andreas Leithner, Lukas Weigl

**Affiliations:** 1Ludwig Boltzmann Institute for Arthritis and Rehabilitation, Thorerstraße 26, 5760 Saalfelden, Austria; 2Ludwig Boltzmann Institute for Arthritis and Rehabilitation, Spitalgasse 23, 1090 Vienna, Austria; 3Department of Orthopedics and Trauma, Medical University of Graz, Auenbruggerplatz 5, 8036 Graz, Austria; 4Karl Chiari Lab for Orthopaedic Biology, Department of Orthopedics and Trauma Surgery, Medical University of Vienna, Waehringer Guertel 18-20, 1090 Vienna, Austria; 5Department of Special Anaesthesia and Pain Therapy, Medical University of Vienna, Waehringer Guertel 18-20, 1090 Vienna, Austria

**Keywords:** mechanotransduction, human chondrocytes, Piezo1, TRPV4, calcium signaling

## Abstract

Osteoarthritis (OA) is the most common degenerative joint disease causing pain and functional limitations. Physical activity as a clinically relevant, effective intervention alleviates pain and promotes joint function. In chondrocytes, perception and transmission of mechanical signals are controlled by mechanosensitive ion channels, whose dysfunction in OA chondrocytes is leading to disease progression. Signaling of mechanosensitive ion channels Piezo/TRPV4 was analyzed by Yoda1/GSK1016790A application and calcium-imaging of Fura-2-loaded chondrocytes. Expression analysis was determined by qPCR and immunofluorescence in healthy vs. OA chondrocytes. Chondrocytes were mechanically stimulated using the Flexcell™ technique. Yoda1 and GSK1016790A caused an increase in intracellular calcium [Ca^2+^]_i_ for Yoda1, depending on extracellularly available Ca^2+^. When used concomitantly, the agonist applied first inhibited the effect of subsequent agonist application, indicating mutual interference between Piezo/TRPV4. Yoda1 increased the expression of metalloproteinases, bone-morphogenic protein, and interleukins in healthy and OA chondrocytes to a different extent. Flexcell™-induced changes in the expression of MMPs and ILs differed from changes induced by Yoda1. We conclude that Piezo1/TRPV4 communicate with each other, an interference that may be impaired in OA chondrocytes. It is important to consider that mechanical stimulation may have different effects on OA depending on its intensity.

## 1. Introduction

Osteoarthritis (OA) is the most common degenerative joint disease that causes functional limitations in addition to pain and has become a major global health problem in recent decades. Considering that the number of OA cases worldwide has more than doubled from 247.51 million in 1990 to 527.81 million in 2019, the consequences of disease-related disabilities may lead to a huge socioeconomic burden in the future [1]. Meta-analyses of studies on patients with knee OA led to the conclusion that physical exercise can function as a clinically relevant, effective intervention and non-drug treatment for reducing pain and improving movement [2,3]. Mechanical loading of joints plays a critical role in maintaining articular cartilage health and function by promoting anabolic cartilage-maintaining processes in chondrocytes. On the contrary, excessive non-physiological mechanical loading is detrimental to cartilage and may lead to an increased risk of developing OA [4].

Mechanical properties of articular cartilage depend to a large extent on the microstructure of the pericellular and extracellular matrix (PCM, ECM) [5,6]. At the molecular level, dynamic compressive loading of cartilage promotes ECM biosynthesis, resulting in improved mechanical properties. Abnormal overload can disrupt ECM homeostasis, trigger catabolic response, lead to tissue damage and promote OA [7,8]. The build-up of the ECM of cartilage during development and its maintenance in healthy tissue is carried out by the metabolic activity of chondrocytes [9,10]. The reciprocal relationship between chondrocytes and the PCM/ECM is based on the ability of chondrocytes to sense physical signals and transduce them into biochemical responses, making them highly mechanosensitive. This conversion of mechanical signals into chemical signals is called mechanotransduction and enables chondrocytes to detect changes in ECM properties [11,12,13]. Based on previous findings, the question arises whether the complex process of mechanotransduction under OA conditions might be altered in a way that cartilage-preserving chondrocytes show different sensitivity to mechanical stimuli due to the disease. Mechanical signals that are positively transduced by healthy cartilage tissue may already be deleterious under OA conditions, leading to a misregulation of matrix metalloproteinase (MMP) and proinflammatory interleukins (ILs) expression.

ECM deformation activates a series of ion channels in chondrocytes. As a consequence, changes in intracellular calcium concentration ([Ca^2+^]_i_) via channel activation are among the most fundamental molecular responses to physical stimuli. [Ca^2+^]_i_ can be increased via Ca^2+^ influx from the extracellular environment along the gradient or the Ca^2+^ release from intracellular stores [14]. Both mechanisms are involved in regulating Ca^2+^ flux in chondrocytes under physical stimulation. In chondrocytes, the perception and transduction of mechanical signals leading to changes in [Ca^2+^]_i_ is orchestrated by mechanosensitive ion channels such as Piezo1, Piezo2, and the transient receptor potential vanilloid type 4 cation channel (TRPV4) [15]. Piezo1 and Piezo2 are mechanosensitive cation channels that allow Ca^2+^ influx also into chondrocytes and act as important mechanotransducers that sense deleterious mechanical stresses (high strains) [16,17]. Regarding chemical channel activation, a small synthetic molecule termed Yoda1 functions as a channel agonist [18] and stabilizes the open Piezo1 channel. It acts as a molecular wedge that facilitates force-induced conformational changes, effectively lowering the mechanical threshold for channel activation [19]. The Ca^2+^-permeable cation channel TRPV4 senses and transmits mechanical signals and osmotic signals in various tissues of the musculoskeletal system, including cartilage, bone, and synovium. The importance of TRPV4 is apparent in patients harboring TRPV4 mutations, which result in the development of a spectrum of skeletal dysplasias and arthropathies [20,21]. The genetic knockout of TRPV4 results in the development of OA and decreased osteoclast function [22]. Furthermore, chemical activation of TRPV4 by the agonist GSK1016790A in the absence of mechanical loading similarly enhanced anabolic and suppressed catabolic gene expression and blocked IL-1β mediated articular cartilage matrix destruction [23]. These findings support the hypothesis that TRPV4-mediated Ca^2+^ signaling plays a central role in the transduction of mechanical signals to support cartilage extracellular matrix maintenance and joint health. Whereas TRPV4-mediated Ca^2+^ appears to support ECM gene expression, the effects of the activation of Piezo1 on the expression of ECM remodeling factors remain to be elucidated.

Numerous studies have shown that Ca^2+^ regulation in conjunction with mechanical stimulation is crucial for the behavior and function of chondrocytes and the balanced assembly of the ECM. From a therapeutic perspective, exercise therapy is an essential component of the OA treatment strategy. Therefore, it is important to characterize the abnormalities in mechanical signal transduction caused by OA to determine the extent to which mechanical stimulation of cartilage may have a positive or negative effect at the molecular level. The work presented here is intended to contribute to the elucidation of changes related to mechanical signal transduction in healthy and diseased chondrocytes.

## 2. Results

### 2.1. Chemical Activation of Piezo1 by Yoda1 in Primary Human OA Chondrocytes

The effect of Yoda1 on primary human OA chondrocytes (pCH-OA) in its function as an agonist of Piezo1 was investigated in terms of channel activation and functionality in the absence of mechanical stimuli. The application of different concentrations of Yoda1 revealed a concentration-dependent increase in [Ca^2+^]_i_ determined in Fura-2 loaded pCH-OA cells via Ca^2+^ imaging (Figure 1a–c). To test whether the increase in [Ca^2+^]_i_ is due to Ca^2+^ influx rather than activation of intracellular Ca^2+^ stores, Ca^2+^ was omitted from the extracellular solution (Ca^2+^ex) during perfusion with 1.0 μM Yoda1. As a result, [Ca^2+^]_i_ decreased significantly from 433.455 ± 42.365 nM [Ca^2+^]_i_ to 282.727 ± 53.543 nM [Ca^2+^]_i_, with the effect reversed when Ca^2+^ was added again (Figure 1a,d). Pretreatment with Thapsigargin, thereby depleting intracellular Ca^2+^ stores, did not abolish the increase in [Ca^2+^]_i_ induced by Yoda1 (Figure 1b). The depletion of stores by Thapsigargin was verified by the response of [Ca^2+^]_i_ in pCH-OA cells to histamine (Figure 1c). Concentration-dependent responses to Yoda1 were neither affected by pretreatment with Thapsigargin nor by the intermediate omission of Ca^2+^ from the extracellular solution (Figure 1e). The experiments show that the observed Yoda1-induced increase in [Ca^2+^]_i_ is primarily caused by Ca^2+^ influx rather than due to the release of Ca^2+^ from intracellular stores. The course of Ca^2+^ changes by Yoda1 is comparable between pCH-OA and healthy chondrocytes (HC).

### 2.2. Interaction and Mutual Influence of GSK1016790A and Yoda1 Effects

We were able to show that the Yoda1 effect in pCH-OA is not due to the release of intracellular calcium and that Yoda1 efficacy diminishes in the absence of extracellular calcium. In view of these results, we also attempted to treat the cells with GSK1016790A (GSK), an agonist for TRPV4, before treating the cells with Yoda1 and vice versa (Figure 2a,b). In order to clarify the influence of OA in this context, we also addressed the question of whether mechanosensitive mechanisms in pCH-OA differ from those in HC cells (chondrocytes from healthy cartilage). Calculation of the correlation between GSK-induced increase in [Ca^2+^]_i_ and the subsequent Yoda1 effect ([Ca^2+^]_iYoda1_ minus ([Ca^2+^]_iGSK_) showed a significant negative correlation in HC (r = −0.615, *p* = 0.043), whereas no correlation was detected in pCH-OA (r = 0.035, *p* = 0.909) (Figure 2c). If one compares the initial response of chondrocytes to Yoda1 with the impact of subsequent GSK treatment, a significant negative correlation in both cell types was detected (HC: r = −0.763, *p* = 0.017; pCH-OA: r = −0.789, *p* = 0.035) (Figure 2d). Accordingly, after a preliminary treatment with GSK, subsequent Yoda1 application showed only a small increase in [Ca^2+^]_i_ (mean ± SEM; HC: 82.981 ± 22.647 nM; pCH-OA: 207.923 ± 66.372 nM). This observed minimal Yoda1 effect differed significantly compared to the initial increase in [Ca^2+^]_i_ by GSK (HC: 711.745 ± 193.648 nM; pCH-OA: 860.538 ± 154.358 nM) (Figure 2e). Reversing the experimental setup showed that treatment with Yoda1 triggered an increase in [Ca^2+^]_i_ (HC: 707.111 ± 163.785 nM; pCH-OA: 1154.500 ± 275.658 nM) that was barely enhanced by GSK. In some cases, the effect of GSK was too small to compensate for the decrease in [Ca^2+^]_i_ seen after the maximum effect of Yoda1, thus resulting in negative values of [Ca^2+^]_i_ in the analysis (mean ± SEM; HC: −81.888 ± 84.644 nM; pCH-OA: −191.714 ± 204.416 nM) (Figure 2e).

The initial treatment with the respective first agonist inhibited the effect of subsequent treatment with the respective second agonist. For Yoda1, a prior GSK treatment reduced the effect to 12.2% in HC and 24.9% in pCH-OA of the original Yoda1 response (Figure 2e). Conversely, pretreatment with Yoda1 reduced and partially prevented the response to GSK as shown by the differences in [Ca^2+^]_i_ in HC (GSK: 486.371 ± 157.844; GSK after Yoda1: −101.875 ± 93.263) and pCH-OA (GSK: 857.000 ± 147.407; GSK after Yoda1: −185.667 ± 241.763), although both cell types responded properly to initial GSK application. From these results, we infer that there may be a close connection between these two types of mechanosensitive ion channels and that this “interaction” may somehow be disrupted under disease conditions.

### 2.3. Yoda1-Induced Increase in [Ca^2+^]_i_ Does Not Depend on the Intracellular Ca^2+^ Level

Based on the above results, we examined whether the differential magnitude of the response to Yoda1 was dependent on the level of [Ca^2+^]_i_ or might be a result of channel interaction in the course of Piezo1 and TRPV4 activation. Changes in [Ca^2+^]_i_ was induced by adding the Ca^2+^-ionophore 4-Br-A23187 to the bath solution. Within the cells tested, 1.0 μM Yoda1 was sufficient to induce an increase in [Ca^2+^]_i_ (Figure 3a), and the application of the Ca^2+^-ionophor only slightly affected the additional effect of Yoda1 (Figure 3b). The measured Yoda1-induced increase after 4-Br-A12387 application (459.417 ± 47.640) was not significantly different compared to Yoda1 measurements alone (517.944 ± 99.731) (Figure 3c), and there was no correlation of the ionophor-induced [Ca^2+^]_i_ with the subsequent[Ca^2+^]_i_ increase by Yoda1 (r = 0.269, *p* = n.s.) (Figure 3d).

### 2.4. Expression of Mechanosensitive Cation Channels in pCH-OA and HC

Subsequently, attention also turned to TRPV4, because-assuming the Yoda1 effect might not depend on the [Ca^2+^]_i_ level-a functional dependence between the two channels (i.e., Piezo1 and TRPV4) seems to be possible. For this reason, we first examined the extent to which the cells expressed mechanosensitive cation channels. Immunofluorescence (IF) studies provided insight into the expression pattern of Piezo1, Piezo2, and TRPV4 in response to Yoda1, both in pCH-OA and HC (Figure 4a). No significant Yoda1-induced change in the expression of mechanosensitive ion channels in comparison to mock-treated controls could be observed. Whereas the expression of all three channels investigated seemed to be slightly increased in pCH-OA compared to HC controls, there was a highly significant 5-fold difference of TRPV4 in pCH-OA (2.735 ± 0.151fold change) compared to HC (0.545 ± 0.053-fold change) in the presence of 0.1 μM Yoda1. Representative images are shown (Figure 4b). To gain new insights into the differences between pCH-OA and HC, agonist-induced channel activation was also compared. There was no difference between the two cell types in their response to Yoda1 (Figure 4c), but the activation of TRPV4 by GSK was significantly different. The GSK effect in pCH-OA (857.000 ± 147.407) was 2.08-fold higher than in HC (411.861 ± 150.300) (Figure 4d). The fact that the GSK effect in pCH-OA was more pronounced than in HC corresponds to the different intensities seen with the IF staining. To confirm these observations at the RNA level, qRT-PCRs were performed, which led to the conclusion that Piezo2 and TRPV4 are higher expressed in pCH-OA than in HC, as shown by the decrease in ΔCT levels (Figure 4e). From these observations, we conclude that there is a difference in the expression of mechanosensitive cation channels between HC and pCH-OA, particularly with respect to TRPV4, where we also found a functional difference between these two cell types.

### 2.5. Yoda 1 Induced Changes in the Expression of MMPs, TIMP-2, BMP-2, Col1A1, and Interleukins

The relative gene expression of MMP1, MMP3, and MMP13 was strongly modulated by Yoda1, whereas for MMP3 and MMP13, the increase was found to be slightly concentration-dependent in HC only (Figure 5a–c). MMP1 expression highly increased in HC by 8.462 ± 2.150 and 449.192 ± 137.043-fold under 1.0 and 3.0 μM Yoda1, respectively, whereas pCH-OA responded to 3 μM Yoda1 with only about one-tenth of the effect than HC (47.011 ± 24.740) (Figure 5a). In a similar manner, MMP3 expression levels elevated by 2.284 ± 0.250, 15.786 ± 3.606, and 504.591 ± 104.410-fold when HC was treated with 0.3, 1.0, and 3.0 μM Yoda1, respectively. MMP3 expression in pCH-OA cells changed only in the presence of 1.0 μM (1.425 ± 0.239) and 3.0 μM (65.418 ± 27.923) Yoda1 (Figure 5b). In comparison, there is an 11.078-fold (1.0 μM Yoda1) and 7.713-fold (3.0 μM) difference in Yoda1-induced MMP3 expression between pCH-OA and HC. Surprisingly, the expression of MMP13 showed opposite regulation comparing the two cell types. In HC cells, MMP13 increased steadily with increasing Yoda1 concentration (1.353 ± 0.088, 1.857 ± 0.1687, 2.985 ± 0.596 for 0.1, 0.3, 1.0 μM Yoda1), whereas in pCH-OA a decrease in MMP13 expression was detected at 1.0 μM Yoda1 (0.772 ± 0.135) (Figure 5c). The tissue inhibitor of metalloproteinase-2 (TIMP-2), an inhibitor of MMPs differentially regulated in chondrocytes with basal levels necessary for the cartilage ECM integrity, was downregulated by Yoda1 in both cell types. At 1 μM Yoda1, TIMP-2 expression was reduced by half in both cell types (HC: 0.523 ± 0.083; pCH-OA: 0.613 ± 0.142) (Figure 5d). Yoda1 affected not only the expression of ECM-modifying enzymes but also the expression of bone morphogenetic protein 2 (BMP-2) by increasing expression to different extents depending on Yoda1 concentration (Figure 5e). While lower concentrations of Yoda1 only slightly increased BMP-2 (0.1 μM: 1.537 ± 0.193 (HC), 2.096 ± 0.291 (pCH-OA); 0.3 μM: 1.884 ± 0.141 (HC), 1.807 ± 0.302 (pCH-OA)), higher concentrations produced a striking highly significant increase (1.0 μM: 18.467 ± 6.277 (HC), 13.240 ± 1.404 (pCH-OA); 3.0 μM: 254.564 ± 88.707 (HC), 71.371 ± 2.819 (pCH-OA)). Notably, at 3.0 μM Yoda1, the increase in HC for BMP-2 was 3.6-fold higher than in pCH-OA. Although no effects on Col2A1 expression were detected, treatment of cells with Yoda resulted in a decrease in Col1A1 expression (Figure 5f): in the presence of 1 μM Yoda1, the decrease was significant in pCH-OA (pCH-OA: 0.510 ± 0.074), whereas under 3.0 μM Yoda1, it was highly significant in both cell types (HC: 0.141 ± 0.026; pCH-OA: 0.172 ± 0.081).

Yoda1 treatment further influenced factors involved in inflammatory processes in chondrocytes. While the gene expression level of interleukin-4 (IL4), which protects mechanically overloaded chondrocytes from degradation, was decreased by 3.0 μM Yoda1 in HC (0.252 ± 0.092), no similar effect was observed in pCH-OA (Figure 6a). On the contrary, the proinflammatory IL6 and IL8 were increased by Yoda1 in both cell types (Figure 6b,c). A significant increase of IL6 was detected at 1.0 μM Yoda1 (HC: 4.957 ± 2.077; pCH-OA: 3.461 ± 1.819). As for the expression of IL8, a marked increase was induced by Yoda1 in HC (0.3 μM: 3.962 ± 1.407; 1.0 μM: 8.868 ± 2.717; 3.0 μM: 24.690 ± 6.480). Expression of IL8 was much more pronounced by higher Yoda1 concentration in pCH-OA (1.0 μM: 3.607 ± 0.547; 3.0 μM: 148.158 ± 15.472). Our results, therefore, show that under OA conditions, the extent of Yoda1-induced changes is lower than in healthy cells, except for interleukins. This could lead to the interpretation that mechanical activation under OA has a positive effect in the absence of inflammation.

### 2.6. Regulators and Transmitters of Mechanical Signals in pCH-OA and HC

To uncover potential differences in the transmission of mechanical signals in diseased compared to healthy cells, relevant targets were analyzed for their fold changes in both cell types (Figure 7a). A 7.0-fold higher expression of integrin α1 as well as a 2.5- and 2.1-fold higher expression of integrin α5 and β1, respectively, were detected compared with the expression level of HC. The expression of TIMP1 was significantly lower in pCH-OA (0.512 ± 0.064), whereas receptors and Ras-related proteins involved in integrin activation and mechanical signaling were slightly higher expressed than in HC (interleukin-6-receptor, IL6R: 1.324 ± 0.270; Ras-related protein RAP2: 1.877 ± 0.345; Discoidin domain receptor tyrosine kinase 2, DDR2: 2.064 ± 0.445).

Application of mechanical stimulation to pCH-OA using the Flexcell™ tension system for 48 h and testing for ECM-modulating enzymes revealed changes in the expression of MMP1 and MMP3 (Figure 7b). The expression of MMP1 changed in the same direction as in cells stimulated with the Yoda1 agonist (1.740 ± 0.268). In contrast, the expression of MMP3 decreased significantly (0.398 ± 0.021). While MMP13 did not seem to be affected and Col1A1 was only insignificantly increased, IL6 was reduced by half under the influence of mechanical stimulation (0.461 ± 0.027).

GTPase isoforms associated with MAPK/ERK signaling and integrin regulation were examined for their expression behavior for the first time. To translate the results to signal transduction associated with mechanical stimulation and for comparison with HC, integrins, the purinergic receptor P2Y2, and downstream regulators were tested to gain initial insight. Cells were cultured under control conditions and by applying mechanical stimulation. Normalization to unstimulated cells showed differences in regulation in HC and pCH-OA. Under the influence of mechanical stimulation, P2Y2 decreased in HC, while no change was detected in pCH-OA (HC: 0.679 ± 0.055; pCH-OA: 1.087 ± 0.126). Src and H-Ras slightly increased under mechanical stimulation in HC (Src: HC: 1.407 ± 0.511; pCH-OA: 1.132 ± 0.123; H-Ras: HC: 1.219 ± 0.032; pCH-OA: 1.026 ± 0.035), whereas K-Ras increased at the same rate in both cell types (Figure 7c). While we observed markedly increased expression of components of the apparatus that detects and transmits mechanical signals, stimulation with the Flexcell™ system only partially reflected the changes in expression also observed under Yoda1. These data may support the view that in OA, sensitivity to certain mechanical forces is altered by overexpression of the relevant proteins. Preliminary studies of downstream signaling triggered by Flexcell™ revealed only vague differences, leading to the interpretation that the type, level, and intensity of mechanical stimulation may play a significant role.

## 3. Discussion

Articular cartilage enables painless, low-friction movement of synovial joints and contains loosely distributed, highly specialized cells (i.e., chondrocytes) embedded in a matrix that imparts remarkable mechanical properties to articular cartilage. Mechanical stimulation, including fluid shear stress, stretching, compression, and cell swelling, as well as decreased mechanical conditions, can alter cellular membrane potential, thereby regulating the metabolism of articular cartilage [24,25]. In damaged cartilage and misdirected chondrocytes (OA), mechanical stimuli can affect cartilage homeostasis and accelerate cartilage degeneration. Although the loss of TRPV4 in chondrocytes did not alter OA pathogenesis after joint destabilization [26], the imbalanced function of mechanosensitive ion channels, such as Piezo1 and TRPV4, has been implicated in causing cartilage damage [27,28]. Related to this, we found that the Ca^2+^ increase triggered by Yoda1 is mainly due to calcium influx from the extracellular space and that the activation of the two mechanosensitive ion channels Piezo1 and TRPV4 are linked to each other. TRPV4 itself has been shown to interact with transmembrane proteins such as integrins and cytoskeletal elements but can also relay a range of environmental signals [29]. In pancreatic cells, it was discovered that stimulation by Piezo1 triggers TRPV4 channel opening, leading to intracellular dysfunction [30]. From our observations, we can conclude that Piezo1 activation inhibits subsequent TRPV4 activation to a greater extent than vice versa. By examining the effect of increased intracellular calcium through the application of calcium ionophores, we were able to exclude the possibility that inhibition of channel activation underlies pure changes in intracellular calcium. We, therefore, postulate that this dependence is due to a direct or indirect interaction of the two channel proteins. The correlation between Yoda1 and GSK after Yoda1 was stronger in pCH-OA cells, which could be explained by the increased expression of mechanosensitive ion channels, particularly TRPV4. It seems likely, therefore, that a situation similar to that in endothelial cells exists, where the effects of shear stress have been shown to be triggered by Piezo1 but to require TRPV4 [31].

As a limitation of this study, however, we need to consider that the 2D culturing models used in this study might not reflect the 3D environment of native articular cartilage. In addition, although we found a mutual influence of Yoda1 and GSK channel activation in that one seems to suppress the other, we cannot exclude that this is also influenced by the different modes of action of the two agonists. Since we assume that the two channels influence each other during activation, either by a direct or indirect mechanism, it is essential for a better understanding to further investigate the contribution of the two ion channels to this effect and also to include this in the context of OA in the next planned experiments.

As a perspective from another angle, it is also likely that OA changes the communication between ion channels leading to altered sensitivity to a mechanical signal. Subsequently, such changes may affect the expression profile of, i.e., MMPs, ILs, and BMP2 in chondrocytes and, thus, modulating the maintenance of the ECM. In this regard, stimulation of OA cells by Yoda1 appeared to downregulate the expression of ECM-modifying enzymes, suggesting that exercise may arrest ECM destruction to some extent. It is also interesting that Yoda1 did not induce BMP2 in pCH-OA cells to the same extent as in healthy cells, meaning that activation of Piezo1 in OA cells appears to result in lower production of BMP2. Notably, serum and synovial fluid BMP2 levels were shown to be associated with radiographic and symptomatic severity of knee OA, allowing BMP2 to serve as an alternative biochemical parameter for determining the disease severity of primary knee OA [32,33]. Because it was shown that serum BMP-2 levels were higher in patients with knee OA than in healthy controls, our observation of a decrease in BMP2 expression in pCH-OA cells under Yoda1 could be interpreted as a protective effect in OA cells. On the other hand, Yoda1 increased the expression of proinflammatory cytokines in healthy and OA cells, suggesting that Yoda1-activated Piezo1 increases inflammation under all circumstances, as also demonstrated in cardiac fibroblasts [34]. Furthermore, Lee et al. 2021 observed an IL1α-mediated inflammatory signaling pathway in articular cartilage cells that upregulates Piezo1 expression [35]. From these perspectives, it is important to weigh whether exercise/exercise therapy was beneficial or more counterproductive in the context of OA, where Piezo1 is considered a sensor of mechanical force and a transducer of that force into biological effects [36].

Our data provide preliminary evidence that there is channel interaction in chondrocytes and that this channel interaction is relaxed in OA cells, possibly due to increased TRPV4 levels leading to changes in the expression of ECM-modulating enzymes and interleukins. Evidence suggests that both Piezo1 and TRPV4 contribute to separate but overlapping mechano-electrical transduction pathways in chondrocytes [37]. While channel activation via Yoda1 appears to be quite intense, mechanical tension on chondrocytes showed similar but only minor effects, suggesting that the expression of relevant proteins is strongly dependent on the intensity of mechanical stimulation. It is also interesting to note that components of the integrin complex are more highly expressed in pCH-OA cells. This observation is equivalent to the cells becoming more sensitive in the diseased state or/and, on the other hand, better able to cope with mechanical stress. Elements of intracellular signal transduction showed only subtle changes, e.g., in Ras expression, so that, in view of the massive changes in expression of some other targets, it seems particularly important to find out what the differences in intracellular signal transduction between OA and healthy cells might be.

## 4. Materials and Methods

### 4.1. Cartilage Samples and Cell Culture

This study was carried out at the Medical University of Graz and the Medical University of Vienna, Austria, in accordance with the ethical standards of the responsible committee (Ethics vote number of the Medical University of Graz: 31-133ex18/19; Ethics vote number of the Medical University of Vienna: 1822/2017) and the Helsinki Declaration. Patients (n = 15) aged 73.4 ± 5.8 years who underwent end-stage knee arthroplasty for knee OA were included and gave their written informed consent. Femoral cartilage was collected intraoperatively and digested using 2 mg/mL collagenase B (Gibco Invitrogen, Carlsbad, CA, USA) in a chondrocyte growth medium at 37 °C for 24 h. After filtration using a 40 μm filter and centrifugation, the primary chondrocytes (pCH-OA) were seeded in cell culture flasks. For pCH-OA, passages 1 and 2 were used for experiments. Human healthy chondrocytes (HC), isolated from a 65-year-old Caucasian male purchased from Cell Application, Inc. (San Diego, CA, USA), were used as healthy control cells up to passage 4, based on experimental experience. To prevent and counteract the dedifferentiation processes of chondrocytes, the chondrocyte-specific growth medium consisting of DMEM/F12, supplemented with 10% fetal bovine serum (FBS), 1% Penicillin-Streptomycin (5.000 U/mL), 1% L-Glutamine, 1% Insulin-Transferrin-Selen, 0.01% TGF-β (10 ng/mL), and 0.01% FGF (10 ng/mL; all Gibco Invitrogen) was used throughout all experiments. Cells were kept in a humidified 5% CO_2_ atmosphere at 37 °C.

### 4.2. Calcium Imaging

[Ca^2+^]_i_ was assessed with the ratiometric fluorescent dye Fura2-AM (Life Technologies, Carlsbad, CA, USA). Chondrocytes were incubated for 35 to 45 min in loading buffer (NT Tyrode’s solution: 5.6 mM glucose, 137 mM NaCl, 5.4 mM KCl, 2.2 mM NaHCO_3_, 1.8 mM CaCl_2_, 1.1 mM MgCl_2_, 0.4 mM H_2_PO_4_, 10 mM HEPES/Na, pH = 7.4) with 3.5–5 μM Fura2/AM; (Molecular Probes) and 0.025% Pluronic) at room temperature. Thapsigargin, GSK1016790A, Yoda1, and 4-Br-A23187 were purchased from Sigma Aldrich and were added to the perfusion solution in the appropriate concentrations. Coverslips were washed and placed into a perfusion chamber of a Nikon Diaphot 300 (Nikon, Tokyo, Japan) fluorescence microscope at 40× magnification. Fluorescence images were taken at excitation wavelengths of 340 and 380 nm with an emission wavelength of 510 nm. Images were recorded with a sample interval of 1 s and analyzed with the Visitech software version 3.1 (Visitron Systems, Puchheim, Germany). Calibration of fluorescence signals to calculate [Ca^2+^]_i_ was performed according to Grynkiewicz et al. and Thomas and Delaville [38,39]. Background subtraction, rationing of images, and calculation of [Ca^2+^]_i_ were performed offline using the Sigma Plot software version 14.5 (Systat Software Inc., San Jose, CA, USA). Histamine (Sigma Aldrich) at concentrations of 1, 3, 10, 30, or 100 μM in NT Tyrode’s solution was applied by a superfusion system with a 7-channel perfusion pipette (List-electronic, Darmstadt, Germany). The system was driven by a valvebank (TSE, Bad Homburg, Germany) with a solution exchange time of less than 500 ms.

### 4.3. Immunofluorescence

Before immunofluorescence staining, untreated and Yoda1-treated chondrocytes were washed with 1x PBS and fixed with 4.5% formaldehyde solution for 15 min. Blocking and permeabilization were performed in one step with 1× PBS containing 1% bovine serum albumin (BSA), 0.3% Triton X-100 (both Sigma Aldrich), and 5% goat serum (DAKO, Glostrup, Denmark) for 15 min. Subsequently, chondrocytes were incubated with the primary antibody diluted 1:100 in staining solution (1× PBS, 1% BSA, 0.1% Triton X-100, 1% goat serum) overnight at 4 °C. The next day, cells were washed and incubated with the second antibody for 1 h. For nuclear staining, 50 μg/mL 4,6-Diamidino-2-Phenylindole solution (DAPI, Thermo Fisher Scientific, Waltham, MA, USA) was added for 10 min. After washing, cells were embedded in Fluoromount-G mounting medium (Invitrogen). Immunofluorescence images were taken using a Diaphot 300 microscope.

### 4.4. Reverse Transcription Polymerase Chain Reaction (RT-PCR)

Total RNA was isolated 24 h after treatment with DMSO (vehicle control, ctrl), 0.1, 0.3, 1, and 3 μM Yoda1 using the RNeasy Mini Kit and DNase-I treatment according to the manufacturer’s manual (Qiagen, Hilden, Germany). Two μg RNA quantified and qualified via optical density measurements were reverse transcribed with the iScript-cDNA Synthesis Kit (BioRad Laboratories Inc., Hercules, CA, USA) using a blend of oligo(dT) and hexamer random primers. Amplification was performed with the SsoAdvanced Universal SYBR Green Supermix (Bio-Rad Laboratories Inc., Hercules, CA, USA) using technical triplicates and measured by the CFX96 Touch (BioRad Laboratories Inc.). The primer sequences, respectively, the QuantiTect primer assay (Qiagen) order numbers for the matrix metalloproteases (MMP1, MMP3, MMP13), the tissue inhibitors of metalloproteinases (TIMP2), the bone morphogenetic protein (BMP-2), type I collagen (Col1A1), type II collagen (Col2A1), interleukins (IL4, IL6, and IL8, the integrin subunits α1, α5, and β1, TIMP-1, the interleukin-6-receptor (IL6R), Ras-related protein (RAP2), Discoidin domain receptor tyrosine kinase 2 (DDR2), K-Ras, H-Ras, Src, and P2Y2 are listed in Table 1. The results were analyzed using the CFX manager software for CFX Real-Time PCR Instruments (Bio-Rad Laboratories Inc., version 3.1) software, and the quantification cycle values (Ct) were exported for statistical analysis. The results with Ct values greater than 32 were excluded from the analysis. Relative quantification of the expression levels was obtained by the ∆∆Ct method, based on the geometric mean of the internal controls ribosomal protein, large, P0 (RPL), and TATA box binding protein (TBP), respectively. The expression level of the target gene was normalized to the reference genes (ΔCt), and the ΔCt of the test sample was normalized to the ΔCt of the control (ΔΔCt). Finally, the expression ratio was calculated with the 2^−ΔΔCt^ method.

### 4.5. Mechanical Stimulation of Chondrocytes

Using the Flexcell FX5K™ Tension System (Flexcell International Corp, Burlington, NC, USA), mechanical cyclic tension straining was applied to the chondrocytes. Six well-pronectin-coated BioFlex culture plates with a flexible silicone membrane allowed deformation of the attached cells. After reaching 70–80% confluence, cells were mechanically stimulated with 15% elongation and 0.2 Hz over a period of 48 h. Immediately afterward, the RNA was isolated, and RT-PCR was performed as indicated above (HC: *n* = 4; pCH-OA: *n* = 8).

### 4.6. Statistical Analysis

Statistical analysis of the data was performed using the standard Microsoft Excel program and the statistical program GraphPad Prism 9. In addition to descriptive statistics, significant differences were determined at the non-categorical scale levels for normally distributed data using Student’s *t*-test in an unpaired experimental design; data without normal distribution were tested for significance compared to controls using the Mann–Whitney test for unpaired data comparison, for paired analysis the Wilcoxon test was used. As pairs, we considered data that describe the different effects within one and the same experiment and, therefore, within one and the same cells. The relationship between the two variables was examined using Pearson’s correlation analysis. Two-sided *p*-values (*p* < 0.0001 ****; *p* < 0.001 ***; *p* < 0.01 **; *p* < 0.05 *) were considered statistically significant.

## 5. Conclusions

From our studies, we suggest the existence of efficient interchannel communication between Piezo1 and TRPV4, which appears to be altered in OA chondrocytes. This alteration and the mechanisms leading to an increase in [Ca^2+^]_i_ could be responsible for an altered expression profile of clinically relevant OA marker genes. In any case, it is clear that strong activation of Piezo1 leads to a decrease in ECM-destroying proteins, whereas inflammatory factors increase. Comparison with mechanical tension also suggests that MMP and IL expression is strongly dependent on the intensity of ion channel activation. Thus, it can be concluded that mechanical stimulation can have different consequences in the treatment of OA and that it should be used carefully.

## Figures and Tables

**Figure 1 ijms-24-07868-f001:**
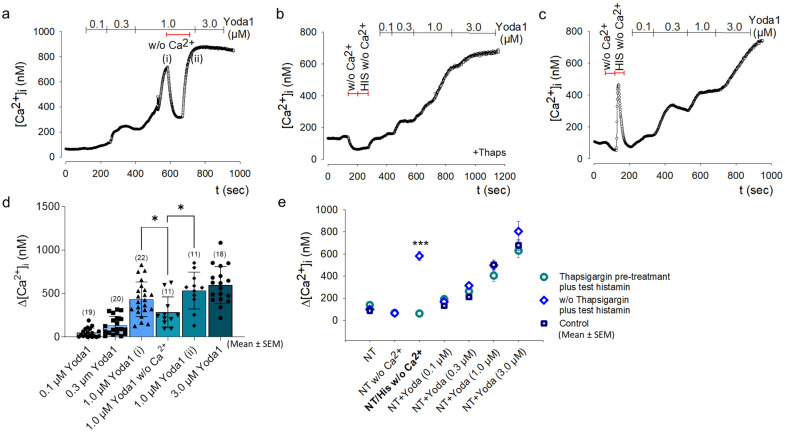
Increase in intracellular Ca^2+^ concentration [Ca^2+^]_i_ by Yoda1 in pCH-OA cells. Response of Fura-2 loaded human primary chondrocytes to 0.1, 0.3, and 1.0 μM Yoda1. Original traces are shown exemplarily from one donor each (**a**–**c**). When Ca^2+^ was omitted from the perfusion solution (NT without, w/o, Ca^2+^), [Ca^2+^]_i_ was reduced (**a**). The effect of Yoda1 was not changed by depleting intracellular Ca^2+^ stores with Thapsigargin (+Thaps), although histamine-induced Ca^2+^ release was completely abolished (HIS w/o Ca^2+^) (**b**). Same measurements as in (**b**) but before Thapsigargin application (**c**). Averaged responses (mean ± SEM) to different concentrations of Yoda1 before and after removal of Ca^2+^ from perfusion solution are shown (**d**). There the different concentrations of Yoda1 correspond to the time course depicted in (**a**), whereas 1.0 μM Yoda1 was added before (i) and after (ii) the period of removal of extracellular Ca^2+^, demonstrating that Ca^2+^ depletion did not influence response to Yoda 1. The corresponding experiments were performed with four different donor cell preparations (preparations of primary OA chondrocytes from four different patients). For each donor cell preparation, the individual treatment conditions were experimentally repeated with different frequencies; this number of experiments is indicated in brackets in the figure in each case. Different treatment options are summarized and compared to each other; no differences except the histamine response in the absence of thapsigargin functioning as a positive control was observed (**e**). Statistically significant differences were calculated unpaired and non-parametric by the Mann–Whitney test; *: *p* < 0.05; ***: *p* < 0.001.

**Figure 2 ijms-24-07868-f002:**
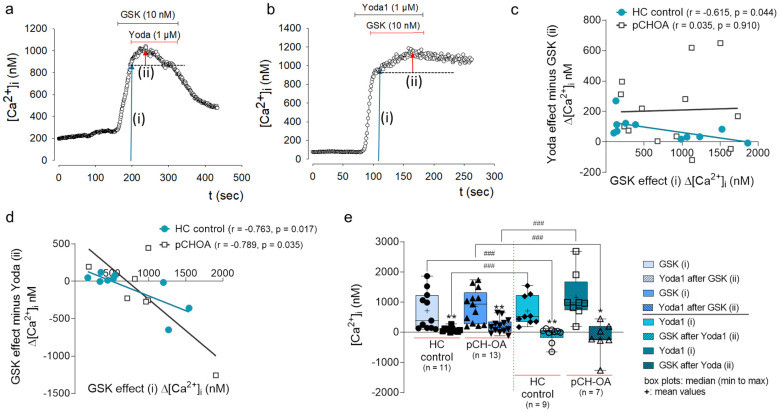
Agonist-dependent activation of Piezo1 and TRPV4. Original traces for sequential activation by GSK and Yoda1 and vice versa in pCH-OA cells; bars in the upper part of the figure mark the addition of the agonists (**a**,**b**); the arrow and letter designation mark the change in [Ca^2+^]_i_ induced by the treatments and therefore the measure points ((i): [Ca^2+^] after GSK, (ii): [Ca^2+^] after additional Yoda1 in (**a**) and (i): [Ca^2+^] after Yoda1, (ii): [Ca^2+^]_i_ after additional GSK in (**b**)). The red arrow symbolizes the hight of the additional effect evoked by Yoda (**a**) or GSK (**b**). For (**c**–**e**), the total amounts of repeated measurements on at least four different donor cells are given in parenthesis. Correlation of GSK effect compared to Yoda1 effect after GSK (HC, n = 11; pCH-OA, n = 13) (**c**). Pearson correlation (r) of reversed treatment, Yoda1 first, GSK in addition thereafter for both cell types is given in (**d**) (HC, n = 9, pCH-OA, n = 7). The boxplot represents the median values (min to max) of the experiments used as a basis for the correlation processes (numbers of replicates are given (n). Significant differences between successive treatments were identified via the Wilcoxon matched-pairs signed rank test (asterisk), whereas for the comparison between the groups, the Mann–Whitney test (hashtag) was used. Significances are given: *: *p* < 0.05, **: *p* < 0.01, ###: *p* < 0.001.

**Figure 3 ijms-24-07868-f003:**
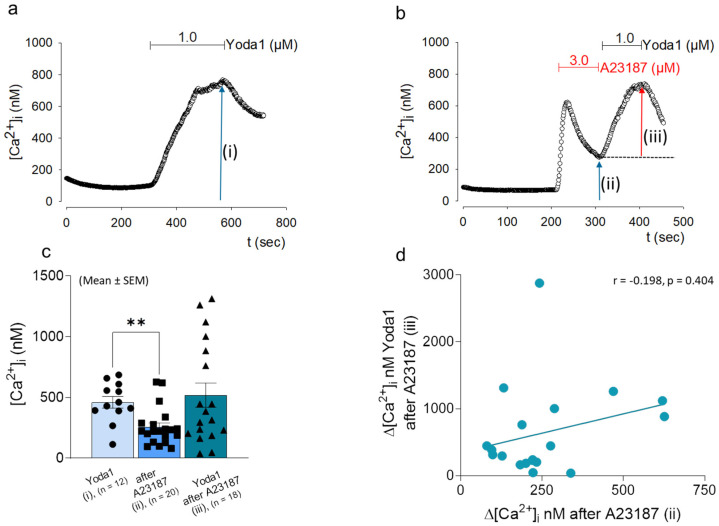
Interplay between Ca^2+^ ionophore A23187 and Yoda1 activation. Time course of the Yoda1 (1.0 μM) induced [Ca^2+^]_i_ (**a**) and the effect of A23187 application (3.0 μM) plus subsequent effect on Yoda1 perfusion (**b**) in pCH-OA cells are given. Arrow and letter designation mark the change in [Ca^2+^]_i_ induced by the treatments, and therefore the measure points (i): [Ca^2+^]_i_ after Yoda1, (ii): [Ca^2+^]_i_ after A23187 and before Yoda1 application, (iii): Yoda1 applied after A23187. The area for A23187 application is marked in red; the bar chart represents the averaged (mean ± SEM) [Ca^2+^]_i_ of the individual treatment regiments; the number of total experiments is given (n). The red arrow symbolizes the hight of the additional effect evoked by Yoda (**b**).The pCH-OA cells used came from eleven different donors and were used with varying frequencies throughout this set of experiments. (**c**). Correlation between values after A23187 application (ii) and subtracted Yoda1 effect (iii) is shown (**d**). Statistical significances calculated by the Mann–Whitney test are given; **: *p* < 0.01.

**Figure 4 ijms-24-07868-f004:**
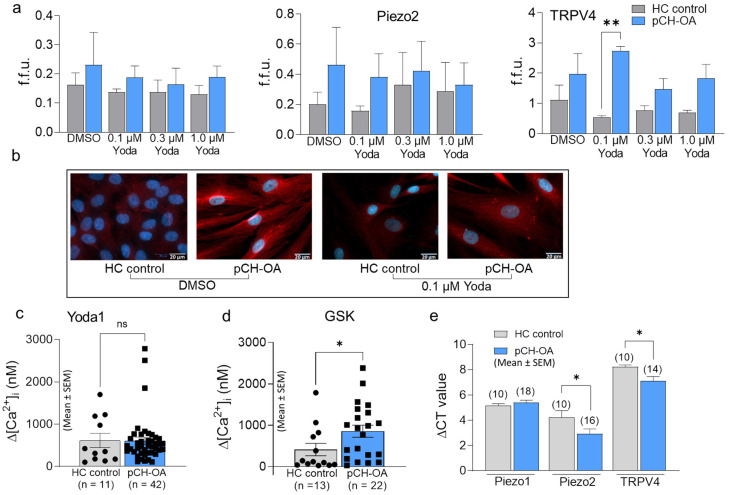
Expression of mechanosensitive channels Piezo1/2 and TPV4 in pCH-OA and HC by immunofluorescence (IF). The bar charts represent the fluorescence forming units (f.f.u.) under control (DMSO; n = 4) and after 24 h of stimulation with 0.1 μM (n = 2), 0.3 μM (n = 4) and 1.0 μM (n = 4) Yoda 1 (**a**). The difference in expression between pCH-OA and HC for TRPV4 under 0.1 μM Yoda1 is statistically significant (**: *p* < 0.01). Representative examples from IF stainings are shown for control conditions (DMSO) and 0.1 μM Yoda1 (**b**). DAPI (4′,6-diamidino-2-phenylindole) (blue) was used to stain nuclei, and anti-rat TRPV4 was used for channel detection (red). Bar charts depict the effect of Yoda1 (**c**) and GSK (**d**) on pCH-OA and HC, total amounts of experiments are given (n). The statistical differences were analyzed via the Mann–Whitney test (*: *p* < 0.05). Under (**e**), mRNA levels of ion channels are presented as ΔCT values (normalized to reference genes); number of total experiments is given in parenthesis. The pCH-OA cells used came from eight different donors and were used with varying frequencies throughout this set of experiments. Significant differences for Piezo2 and TPV4 in pCH-OA cells when compared to HC are given and were calculated by Student’s *t*-test; *: *p* < 0.05.

**Figure 5 ijms-24-07868-f005:**
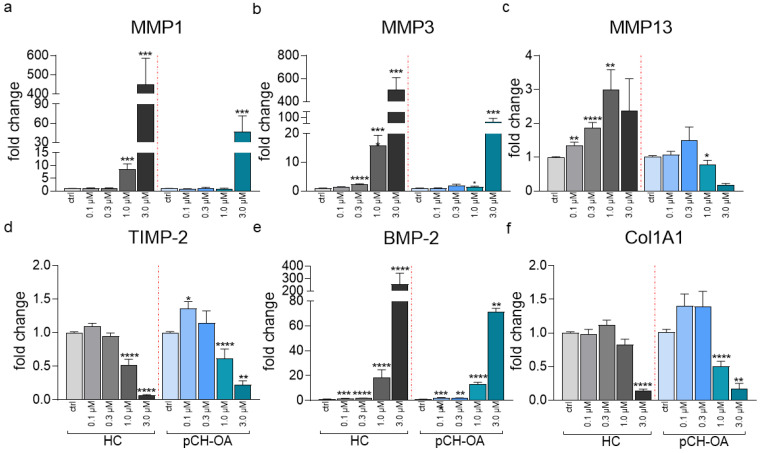
Relative gene expression of the matrix metalloproteinases MMP1 (**a**), MMP3 (**b**), MMP13 (**c**), the tissue inhibitor of metalloproteinase-2 (TIMP-2) (**d**), the bone morphogenetic protein 2 (BMP-2) (**e**), and Collagen type I (Col1A1) (**f**) under the influence of 0 (ctrl, DMSO) and 0.1–3.0 μM Yoda1 is given as mean ± SEM. The changes shown are normalized to DMSO values given as ctrl (vehicle controls). Control healthy (HC, bars greyscale) and OA chondrocytes (pCH-OA, bars blue scale) are shown. The experiments were performed in duplicates, number of donors is given in parenthesis: MMP1 (n = 9), MMP3 (n = 8), MMP13 (n = 9), TIMP-2 (n = 6), BMP-2 (n = 6) and Col1A1 (n = 5). Statistical significance levels were calculated via the Mann–Whitney test; *: *p* < 0.05, **: *p* < 0.01, ***: *p* < 0.001, ****: *p* < 0.0001.

**Figure 6 ijms-24-07868-f006:**
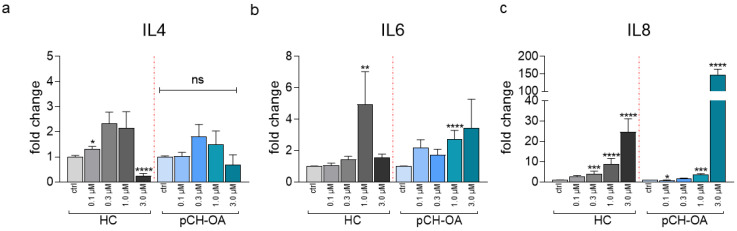
Relative gene expression of interleukins IL4 (**a**), IL6 (**b**), and IL (**c**) under the influence of 0 (ctrl, DMSO) and 0.1–3.0 μM Yoda1 as mean ± SEM is given. Presented fold changes are normalized to DMSO values; control healthy (HC, bars greyscale) and OA chondrocytes (pCH-OA, bars blue scale) are shown. The experiments were performed in duplicates, number of donors is given in parenthesis: IL4 (n = 7), IL-6 (n = 8), and IL-8 (n = 8). Statistical significance levels were calculated via the Mann–Whitney test; *: *p* < 0.05, **: *p* < 0.01, ***: *p* < 0.001, ****: *p* < 0.0001.

**Figure 7 ijms-24-07868-f007:**
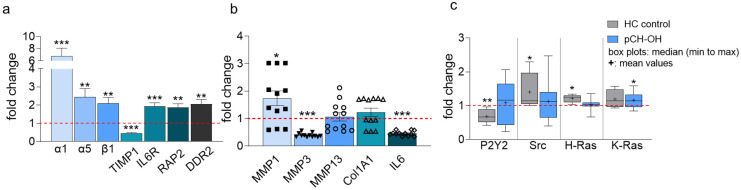
Relative gene expression of the integrin subunits α1, α5, and β1 as well as tissue inhibitor of metalloproteinase-1 (TIMP-1), interleukin-6-receptor (IL6R), ras-related protein Rap-2a (RAP2), and discoidin domain-containing receptor 2 (DDR2) of pCH-OA normalized to HC control cells is shown as mean ± SEM (**a**); HC experimental replicates/pCH-OA experimental replicates (three experimental replicates per pCH-OA donor) were 18/18/(donor = 6) for all targets in (**a**) except IL6R (18/6/(donor = 2)) (**b**) Flexcell™ tension induced expression profile in pCH-OA cells; matrix metalloproteinases MMP1, MMP3, MMP13, collagen I (Col1A1) and the interleukin 6 (IL6) are shown as mean ± SEM; HC experimental replicates/pCH-OA experimental replicates (three experimental replicates per pCH-OA donor) were 12/12/(donor = 4) for all targets in (**b**). (**c**) The box plot given as median (min to max) represents Flexcell™ tension induced changes with respect to the purinergic receptor (P2Y2), Src kinase, H-ras, and K-ras expression profile in healthy (HC) cells compared to pCH-OA cells. The total number of replicates for HC was n = 12, and for pCH-OA, n = 24, where pCH-OA donors were n = 6. For (**a**,**c**), statistical analyses were performed by using Student’s *t*-test, and for (**b**), the Mann–Whitney test was applied. *: *p* < 0.05, **: *p* < 0.01, ***: *p* < 0.001.

**Table 1 ijms-24-07868-t001:** List of qRT-PCR primers with respective 5′–3′ sequence. Primers were designed by NCBI primer blast, checked for alignment, and purchased via Eurogentec or QuantiTect primer assays (QT) were ordered from QIAGEN.

Target	Primer	Sequence 5′–3′
Piezo1	fw	CAT CTT GGT GGT CTC CTC TGT CT
	rev	CTG GCA TCC ACA TCC CTC TCA TC
Piezo2	fw	GCC CAA CAA AGC CAG TTG AA
	rev	GGG CTG ATG GTC CAC AAA GA
TRPV4	QT	00077217
MMP1	fw	CTG TTC AGG GAC AGA ATG TGC T
	rev	TCG ATA TGC TTC ACA GTT CTA GGG
MMP3	fw	TTT TGG CCA TCT CTT CCT TCA
	rev	TGT GGA TGC CTC TTG GGT ATC
MMP13	fw	TCC TCT TCT TGA GCT GGA CTC ATT
	rev	CGC TCT GCA AAC TGG AGG TC
TIMP-2	fw	ACA GGC GTT TTG CAA TGC A
	rev	GGG TTG CCA TAA ATG TCG TTT C
TIMP-1	QT	00084168
BMP-2	QT	00012544
Col1A1	QT	00037793
Col2A1	QT	00049518
IL6	QT	00083720
IL8	QT	00000322
IL4	fw	ATC TTT GCT GCC TCC AAG AAC AC
	rev	GTA GAA CTG CCG GAG CAC AG
N-Ras	fw	CCC GGC TGT GGT CCT AAA TC
	rev	TCC AAC CAC CAC CAG TTT GT
K-Ras	fw	TGT TCA CAA AGG TTT TGT TCT C
	rev	CCT TAT AAT AGT TTC CAT TGC CTT G
Src	fw	TGT TCG GAG GCT TCA ACT CC
	rev	CCA CCA GTC TCC CTC TGT GT
P2Y2	fw	CCG CTC GCT GGA CCT CAG CTG
	rev	CTC ACT GCT GCC CAA CAC ATC
Integrin α1	QT	00093723
Integrin α5	QT	00080871
Integrin β1	QT	00068124
RAP2	QT	00228039
DDR2	QT	00047481
B2M	QT	00088935
TBP	QT	00000721
RPLP0	QT	00075012

## Data Availability

Not applicable.

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
