# Peer review of "Activation of the Mechanosensitive Ion Channels Piezo1 and TRPV4 in Primary Human Healthy and Osteoarthritic Chondrocytes Exhibits Ion Channel Crosstalk and Modulates Gene Expression"

_ijms, 2023, doi:10.3390/ijms24097868_

Round 1
Reviewer 1 Report
The authors described te relationship between Piezo/TRPV4 and OA chondrocyte behaivior.
The authors concluded in the manuscript that, even though Yoda1 activation of Piezo decreased MMPs and increased inflammatory factors,
these effects were only partially detectable under flexcell treatment. Therefore, it is important to consider that extent of mechanical stimulation may have different effects on OA.
However, in J Med Chem 2019, 62, 3, 1468-1483, TRPV4 stimulation itself can modulate chondrogenic differentiation.
And also, GSK compounds in their patent could use the interuption of cartilage degradation.
The difference between both effect could be derived from time course of compound stimulation and also cultivation into 3D or 2D conditions.
I think under the authors conditions, I would not doubt the experimental results.
In discussion section, I recommend that the authors should describe the differece of above compound's effects.
Then, the authors should delineate synergy effects of Piezo and TRPV4. Without the discussion of each protein contribution, it is hard to reach into the conclusion of synergy effect of OA chondrocyte behavior.
When the authors will doublecheck above things, the manuscript may be published in the journal.
Author Response
The authors described the relationship between Piezo/TRPV4 and OA chondrocyte behavior. The authors concluded in the manuscript that, even though Yoda1 activation of Piezo decreased MMPs and increased inflammatory factors, these effects were only partially detectable under flexcell treatment. Therefore, it is important to consider that extent of mechanical stimulation may have different effects on OA.
However, in J Med Chem 2019, 62, 3, 1468-1483, TRPV4 stimulation itself can modulate chondrogenic differentiation.
And also, GSK compounds in their patent could use the interruption of cartilage degradation. The difference between both effects could be derived from time course of compound stimulation and also cultivation into 3D or 2D conditions.
I think under the author’s conditions, I would not doubt the experimental results.
In discussion section, I recommend that the authors should describe the difference of above compound's effects.
Authors reply:
Dear reviewer, thank you very much for these valuable comments, which may be very important for our discussion. I agree, it could also be that the observed effect might be influenced by the time course of stimulation by Yoda1 and/or GSK and by cultivation under 3D or 2D conditions.
First, we can say that because of the use of primary chondrocytes in very early passages, from passage 0 to a maximum of passage 2, and because of our experience with the use of specific growth media to prevent dedifferentiation of chondrocytes, we tend to interpret the observed effects as results of channel-modulating effects of the compounds and not as results of intervention of the compounds in differentiation and or dedifferentiation processes. Unfortunately, we are also not able to exclude a possible influence when the cells are grown "only" in 2 D. However, experience has shown that 3 D of chondrocytes may not change mechanistically but rather improves observed effects. For the future it is clear that it will be necessary to consider the 3 D cultures for research within this field. Optimistic statement with one problem – 3 D works perfectly well for the expression studies, but in the context of our real-time measurements related to changes in intracellular calcium, as well as for the treatment through Flexcell, 3D it is difficult to implement, but not impossible.
The reference to the different mechanisms of action of the substances is very justified and could of course lead to a temporal shift in the effects.
According to the literature, the mechanism by which Yoda1 activates the channel is not yet well understood. Yoda1 is described to act like a molecular wedge, facilitating conformational changes of the channel protein, effectively lowering the mechanical threshold for channel activation. (A mechanism for the activation of the mechanosensitive Piezo1 channel by the small molecule Yoda1 | Nature Communications)
From the literature, it is also discussed that GSK affects the availability of the channels by also increasing their expression.
Both modes of action, in their difference, may influence or cause our effects, but the observed lack of enhancement of the effect by Yoda1 plus GSK 1016790A and vice versa and the consequent decay of the effect can hardly be explained by a possible delayed effect/different mode of action of Yoda1 and/or GSK 1016790A. However, we agree with the reviewer that these aspects should not be left out of sight.
Then, the authors should delineate synergy effects of Piezo and TRPV4. Without the discussion of each protein contribution, it is hard to reach into the conclusion of synergy effect of OA chondrocyte behavior
Authors reply:
We definitely agree with the reviewer's point of view. From our results, we can only conclude that because we can only detect small additional effects by Yoda1 and GSK1016790A treatment, and rather one effect cancels or reduces the other, we assume that there is communication between the two channels. This communication may occur via a direct or indirect interaction. In this preliminary work on mechanosensitive ion channels in chondrocytes, it was important to show differences between healthy and diseased cells and to find a way to explain why cells under OA are more sensitive to mechanical stress. Further intensive studies in the direction of altered communication between these two channels may shed light on this very problem.
From what we learned from this reviewer we added the following comment to the discussion part:
But we also have to consider that based on our observations, we cannot exclude the possibility that culturing the cells only in 2D instead of 3D may influence our results. In addition, although we found a mutual influence of Yoda1 and GSK channel activation in that one seems to suppress the other, we cannot exclude that this is also influenced by the different mode of action of the two agonists. Since we assume that the two channels influence each other during activation, either by a direct or indirect mechanism, it is essential for a better understanding to further investigate the contribution of the two ion channels to this effect and also to include this in the context of OA in the next planned experiments.
We thank Reviewer 1 for her/his time and efforts.
The authors described the relationship between Piezo/TRPV4 and OA chondrocyte behavior. The authors concluded in the manuscript that, even though Yoda1 activation of Piezo decreased MMPs and increased inflammatory factors, these effects were only partially detectable under flexcell treatment. Therefore, it is important to consider that extent of mechanical stimulation may have different effects on OA.
However, in J Med Chem 2019, 62, 3, 1468-1483, TRPV4 stimulation itself can modulate chondrogenic differentiation.
And also, GSK compounds in their patent could use the interruption of cartilage degradation. The difference between both effects could be derived from time course of compound stimulation and also cultivation into 3D or 2D conditions.
I think under the author’s conditions, I would not doubt the experimental results.
In discussion section, I recommend that the authors should describe the difference of above compound's effects.
Authors reply:
Dear reviewer, thank you very much for these valuable comments, which may be very important for our discussion. I agree, it could also be that the observed effect might be influenced by the time course of stimulation by Yoda1 and/or GSK and by cultivation under 3D or 2D conditions.
First, we can say that because of the use of primary chondrocytes in very early passages, from passage 0 to a maximum of passage 2, and because of our experience with the use of specific growth media to prevent dedifferentiation of chondrocytes, we tend to interpret the observed effects as results of channel-modulating effects of the compounds and not as results of intervention of the compounds in differentiation and or dedifferentiation processes. Unfortunately, we are also not able to exclude a possible influence when the cells are grown "only" in 2 D. However, experience has shown that 3 D of chondrocytes may not change mechanistically but rather improves observed effects. For the future it is clear that it will be necessary to consider the 3 D cultures for research within this field. Optimistic statement with one problem – 3 D works perfectly well for the expression studies, but in the context of our real-time measurements related to changes in intracellular calcium, as well as for the treatment through Flexcell, 3D it is difficult to implement, but not impossible.
The reference to the different mechanisms of action of the substances is very justified and could of course lead to a temporal shift in the effects.
According to the literature, the mechanism by which Yoda1 activates the channel is not yet well understood. Yoda1 is described to act like a molecular wedge, facilitating conformational changes of the channel protein, effectively lowering the mechanical threshold for channel activation. (A mechanism for the activation of the mechanosensitive Piezo1 channel by the small molecule Yoda1 | Nature Communications)
From the literature, it is also discussed that GSK affects the availability of the channels by also increasing their expression.
Both modes of action, in their difference, may influence or cause our effects, but the observed lack of enhancement of the effect by Yoda1 plus GSK 1016790A and vice versa and the consequent decay of the effect can hardly be explained by a possible delayed effect/different mode of action of Yoda1 and/or GSK 1016790A. However, we agree with the reviewer that these aspects should not be left out of sight.
Then, the authors should delineate synergy effects of Piezo and TRPV4. Without the discussion of each protein contribution, it is hard to reach into the conclusion of synergy effect of OA chondrocyte behavior
Authors reply:
We definitely agree with the reviewer's point of view. From our results, we can only conclude that because we can only detect small additional effects by Yoda1 and GSK1016790A treatment, and rather one effect cancels or reduces the other, we assume that there is communication between the two channels. This communication may occur via a direct or indirect interaction. In this preliminary work on mechanosensitive ion channels in chondrocytes, it was important to show differences between healthy and diseased cells and to find a way to explain why cells under OA are more sensitive to mechanical stress. Further intensive studies in the direction of altered communication between these two channels may shed light on this very problem.
From what we learned from this reviewer we added the following comment to the discussion part:
But we also have to consider that based on our observations, we cannot exclude the possibility that culturing the cells only in 2D instead of 3D may influence our results. In addition, although we found a mutual influence of Yoda1 and GSK channel activation in that one seems to suppress the other, we cannot exclude that this is also influenced by the different mode of action of the two agonists. Since we assume that the two channels influence each other during activation, either by a direct or indirect mechanism, it is essential for a better understanding to further investigate the contribution of the two ion channels to this effect and also to include this in the context of OA in the next planned experiments.
We thank Reviewer 1 for her/his time and efforts.
Reviewer 2 Report
The study aimed to investigate the OA-induced changes in mechanical signal transduction in human chondrocytes by analysing intracellular Ca++ concentrations mediated by ion channels Piezo/TRPV4 in responses to Yoda1/GSK as well as cell stretching. Although the study addresses an interesting question, unfortunately, the study suffers several flaws and a lack of clarity, including the experimental design, data presentation/ interpretation, etc. In addition, the manuscript is poorly written. There is a lack of mechanistic study with a logical flow and as a consequence, a coherent story failed to emerge from this study. The conclusions drawn from this study are vague. Even the title does not reflect the focus of the study on OA chondrocytes.
Clarity issue, eg:
Line 100: “The work presented here will help elucidate the OA-induced changes associated with mechanical signal transduction in healthy and diseased cartilage cells.”
Line 136: what are the data shown in Figures 2a, 2b? Are the data generated in healthy cells or OA cells? The data presented in ‘e’ doesn’t look like ‘±SEM’.
Pg477-478: “After reaching 70-80% confluence, cells were treated with 100 nM respectively 1 μM of the mechanosensitive and stretch-activated ion channel inhibitor GsMTx-4 (Abcam, Cambridge, UK)…” This sentence doesn’t make sense.
It is not clear in many places what cell lines were used in studies, eg. Fig2 a&b. Fig3, etc. What are the values presented in Fig3c? Are they the technical replicates in one cell line or biological replicates from different cell lines in OA?
Fig4: What channel protein is labelled in Fig4b? It is sceptical for the data shown in Fig4a TRPV4. Why not show the actual images? What does the ‘n=2-4’ mean? Why did the amounts of experiments vary so much between cell lines and ion channels? It is highly sceptical that these experiments were performed at different times for each cell line or whether the data are directly comparable.
Fig5&6: it is not clear whether the p values indicate comparisons between which samples. Are there any differences between HC and OA cells, for the sake of study?
The issue about the cell lines:
The study used only one normal cell line which is not enough. Usually, it requires at least two cell lines in the study. It was mentioned that 15 patient cell lines were used (Line 406), but there is no description/ info about how many cell lines were used in each experiment/figure. A table with detailed clinical patient info and cell lines used for each experiment/figure should be added to help the readers to see the details.
Experimental design/ Data interpretation issue:
In 2.1/Figure 1, the study only analysed human OA chondrocytes. There is a lack of healthy control cells. Additionally, no info about how many cell lines were used in this study and how many biological replicates were performed and presented, etc.
Line 136: as mentioned above, there is a lack of rationale or justification for why these experiments need to be done (Figure 2).
Fig7 shows the analysis of the gene expression. What are the protein expression, especially for the signalling molecules, as well as their phosphorylation status?
Overall, this manuscript needs to be subjected to major revision through additional experiments and data presentation and re-written.
Author Response
The study aimed to investigate the OA-induced changes in mechanical signal transduction in human chondrocytes by analyzing intracellular Ca++ concentrations mediated by ion channels Piezo/TRPV4 in responses to Yoda1/GSK as well as cell stretching. Although the study addresses an interesting question, unfortunately, the study suffers several flaws and a lack of clarity, including the experimental design, data presentation/ interpretation, etc. In addition, the manuscript is poorly written. There is a lack of mechanistic study with a logical flow and as a consequence, a coherent story failed to emerge from this study. The conclusions drawn from this study are vague.
Even the title does not reflect the focus of the study on OA chondrocytes.
Authors reply: we changed the title to connect this work with OA chondrocytes
Clarity issue, eg:
Line 100: “The work presented here will help elucidate the OA-induced changes associated with mechanical signal transduction in healthy and diseased cartilage cells.”
Authors reply: we changed the sentence for clarification to: “The work presented here is intended to contribute to the elucidation of changes related to mechanical signal transduction in healthy and diseased cartilage cells”.
Line 136: what are the data shown in Figures 2a, 2b? Are the data generated in healthy cells or OA cells? The data presented in ‘e’ doesn’t look like ‘±SEM’.
Authors reply:
- 2. a) and b): figures show representative original traces illustrating the progression of Ca2+ changes as a result of treatment of cells from one individual. These experiments, which served as the starting point for the project, were performed only on pCH-OA cells to fundamentally clarify the mechanisms of calcium stimulation by Yoda1.
- Are the data generated in healthy cells or OA cells? These original traces represent measurements on OA cells.
- The data presented in ‘e’ doesn’t look like ‘±SEM’: we apologize for the incomplete data - these are median values (min to max) - we have now noted this in the figure.
Pg477-478: “After reaching 70-80% confluence, cells were treated with 100 nM respectively 1 μM of the mechanosensitive and stretch-activated ion channel inhibitor GsMTx-4 (Abcam, Cambridge, UK)…” This sentence doesn’t make sense.
Authors reply:
Unfortunately, we obviously made a mistake there, we have deleted this sentence.
It is not clear in many places what cell lines were used in studies, eg. Fig2 a&b. Fig3, etc. What are the values presented in Fig3c? Are they the technical replicates in one cell line or biological replicates from different cell lines in OA?
Authors reply:
According to the editor's criticism and suggestions for improvement, which were absolutely justified and have been worked on in detail by us, these points have already been corrected, supplemented and edited (see letter to the editor plus corrected manuscript).
Fig4:
- What channel protein is labelled in Fig4b?
Authors reply: It is TRPV4 and this omission of the description was corrected
- It is sceptical for the data shown in Fig4a TRPV4. Why not show the actual images? Authors reply: The actual images for TRPV4 are given in Fig4b
- What does the ‘n=2-4’ mean?
Authors reply: This is the number of IF experiments - Fig. 4a represents the results of immunofluorescence experiments and in terms of the experimental setup has nothing to do with the expression analysis regarding ion channel expression in Fig. 4e. One is IF and the other is RNA isolation, so we used a different setting, new cells, and a different number of experiments.
- Why did the amounts of experiments vary so much between cell lines and ion channels?
Authors reply: Because these are two different settings for studying expression – qPCR analysis the amount of RNA and via the IF one is able to detect the mature protein. Both methods aim at the expression of the targeted protein.
- It is highly sceptical that these experiments were performed at different times for each cell line or whether the data are directly comparable.
Authors reply: I apologize but I do not understand this point? The IFs were performed within different experiments at different time points with different donor cells; experiments for qPCR were performed one after the other – repeated with 5 to 8 different donor pCH-OA cells.
Fig5&6: it is not clear whether the p values indicate comparisons between which samples. Are there any differences between HC and OA cells, for the sake of study?
Authors reply: The p-values indicate the comparison with the control labeled with ctrl in each series of experiments, as also indicated in the figure legend. As described in the Results section, there are differences between HC and pCH-OA cells suggesting that activation by Yoda1, and thus mimicked mechanical stimulation, alters the expression of matrix-destroying proteins as well as inflammatory components. Therefore, it is of great importance for the sake of this study to show how and in the context of which mechanical signal pCH-OA cells respond to activation by mechanical stimulation, once again pointing to the importance of elucidating molecular changes under OA.
The issue about the cell lines:
- The study used only one normal cell line which is not enough.
Authors reply: The cells we used were human primary cells and not cell lines. For control purposes, it is notoriously difficult to obtain healthy chondrocytes, so one of the few options is to buy them. We tried to perform the experiments with these cells also in appropriate replicates - that one patient in this case is a bit little is clear to me but otherwise healthy controls are almost not possible.
- Usually, it requires at least two cell lines in the study. It was mentioned that 15 patient cell lines were used (Line 406), but there is no description/ info about how many cell lines were used in each experiment/figure.
Authors reply: We really apologize for this incompleteness and have made up for it in all results, as the editor was kind enough to point out point by point where something is missing.
- A table with detailed clinical patient info and cell lines used for each experiment/figure should be added to help the readers to see the details.
Authors reply: Unfortunately, such information is not provided, since we receive the samples from the operating room only with a number and thus anonymized as is usual in Austria and also corresponds to the concept of the Ethics Committee.
Experimental design/ Data interpretation issue: In 2.1/Figure 1, the study only analysed human OA chondrocytes. There is a lack of healthy control cells. Additionally, no info about how many cell lines were used in this study and how many biological replicates were performed and presented, etc.
Authors reply: Information about replicates, donors and the fact that pCH-OA cells have been used has been added to the manuscript due to the editor’s request. An explanation why using pCH-OA cells has been added.
Line 136: as mentioned above, there is a lack of rationale or justification for why these experiments need to be done (Figure 2).
Authors reply: With the help of this experimental set-up, it could be shown that there is obviously a connection between the activation of Piezo1 and TRPV4. Since it is postulated in the literature that these two channels are involved in the sensitivity to mechanical stress and it is discussed that there might be an imbalance here under OA, this experiment was aimed at characterizing initial physiological connections, i.e. connections on a cellular level.
Fig7 shows the analysis of the gene expression. What are the protein expression, especially for the signalling molecules, as well as their phosphorylation status?
Authors reply: Thank you very much for your valuable comments! These are preliminary experiments to find indications of changes in different signaling pathways/directions to serve as a basis for further investigations, especially for planned experiments regarding protein phosphorylation.
Overall, this manuscript needs to be subjected to major revision through additional experiments and data presentation and re-written.
We thank Reviewer 2 for her/his time and efforts.
The study aimed to investigate the OA-induced changes in mechanical signal transduction in human chondrocytes by analyzing intracellular Ca++ concentrations mediated by ion channels Piezo/TRPV4 in responses to Yoda1/GSK as well as cell stretching. Although the study addresses an interesting question, unfortunately, the study suffers several flaws and a lack of clarity, including the experimental design, data presentation/ interpretation, etc. In addition, the manuscript is poorly written. There is a lack of mechanistic study with a logical flow and as a consequence, a coherent story failed to emerge from this study. The conclusions drawn from this study are vague.
Even the title does not reflect the focus of the study on OA chondrocytes.
Authors reply: we changed the title to connect this work with OA chondrocytes
Clarity issue, eg:
Line 100: “The work presented here will help elucidate the OA-induced changes associated with mechanical signal transduction in healthy and diseased cartilage cells.”
Authors reply: we changed the sentence for clarification to: “The work presented here is intended to contribute to the elucidation of changes related to mechanical signal transduction in healthy and diseased cartilage cells”.
Line 136: what are the data shown in Figures 2a, 2b? Are the data generated in healthy cells or OA cells? The data presented in ‘e’ doesn’t look like ‘±SEM’.
Authors reply:
- 2. a) and b): figures show representative original traces illustrating the progression of Ca2+ changes as a result of treatment of cells from one individual. These experiments, which served as the starting point for the project, were performed only on pCH-OA cells to fundamentally clarify the mechanisms of calcium stimulation by Yoda1.
- Are the data generated in healthy cells or OA cells? These original traces represent measurements on OA cells.
- The data presented in ‘e’ doesn’t look like ‘±SEM’: we apologize for the incomplete data - these are median values (min to max) - we have now noted this in the figure.
Pg477-478: “After reaching 70-80% confluence, cells were treated with 100 nM respectively 1 μM of the mechanosensitive and stretch-activated ion channel inhibitor GsMTx-4 (Abcam, Cambridge, UK)…” This sentence doesn’t make sense.
Authors reply:
Unfortunately, we obviously made a mistake there, we have deleted this sentence.
It is not clear in many places what cell lines were used in studies, eg. Fig2 a&b. Fig3, etc. What are the values presented in Fig3c? Are they the technical replicates in one cell line or biological replicates from different cell lines in OA?
Authors reply:
According to the editor's criticism and suggestions for improvement, which were absolutely justified and have been worked on in detail by us, these points have already been corrected, supplemented and edited (see letter to the editor plus corrected manuscript).
Fig4:
- What channel protein is labelled in Fig4b?
Authors reply: It is TRPV4 and this omission of the description was corrected
- It is sceptical for the data shown in Fig4a TRPV4. Why not show the actual images? Authors reply: The actual images for TRPV4 are given in Fig4b
- What does the ‘n=2-4’ mean?
Authors reply: This is the number of IF experiments - Fig. 4a represents the results of immunofluorescence experiments and in terms of the experimental setup has nothing to do with the expression analysis regarding ion channel expression in Fig. 4e. One is IF and the other is RNA isolation, so we used a different setting, new cells, and a different number of experiments.
- Why did the amounts of experiments vary so much between cell lines and ion channels?
Authors reply: Because these are two different settings for studying expression – qPCR analysis the amount of RNA and via the IF one is able to detect the mature protein. Both methods aim at the expression of the targeted protein.
- It is highly sceptical that these experiments were performed at different times for each cell line or whether the data are directly comparable.
Authors reply: I apologize but I do not understand this point? The IFs were performed within different experiments at different time points with different donor cells; experiments for qPCR were performed one after the other – repeated with 5 to 8 different donor pCH-OA cells.
Fig5&6: it is not clear whether the p values indicate comparisons between which samples. Are there any differences between HC and OA cells, for the sake of study?
Authors reply: The p-values indicate the comparison with the control labeled with ctrl in each series of experiments, as also indicated in the figure legend. As described in the Results section, there are differences between HC and pCH-OA cells suggesting that activation by Yoda1, and thus mimicked mechanical stimulation, alters the expression of matrix-destroying proteins as well as inflammatory components. Therefore, it is of great importance for the sake of this study to show how and in the context of which mechanical signal pCH-OA cells respond to activation by mechanical stimulation, once again pointing to the importance of elucidating molecular changes under OA.
The issue about the cell lines:
- The study used only one normal cell line which is not enough.
Authors reply: The cells we used were human primary cells and not cell lines. For control purposes, it is notoriously difficult to obtain healthy chondrocytes, so one of the few options is to buy them. We tried to perform the experiments with these cells also in appropriate replicates - that one patient in this case is a bit little is clear to me but otherwise healthy controls are almost not possible.
- Usually, it requires at least two cell lines in the study. It was mentioned that 15 patient cell lines were used (Line 406), but there is no description/ info about how many cell lines were used in each experiment/figure.
Authors reply: We really apologize for this incompleteness and have made up for it in all results, as the editor was kind enough to point out point by point where something is missing.
- A table with detailed clinical patient info and cell lines used for each experiment/figure should be added to help the readers to see the details.
Authors reply: Unfortunately, such information is not provided, since we receive the samples from the operating room only with a number and thus anonymized as is usual in Austria and also corresponds to the concept of the Ethics Committee.
Experimental design/ Data interpretation issue: In 2.1/Figure 1, the study only analysed human OA chondrocytes. There is a lack of healthy control cells. Additionally, no info about how many cell lines were used in this study and how many biological replicates were performed and presented, etc.
Authors reply: Information about replicates, donors and the fact that pCH-OA cells have been used has been added to the manuscript due to the editor’s request. An explanation why using pCH-OA cells has been added.
Line 136: as mentioned above, there is a lack of rationale or justification for why these experiments need to be done (Figure 2).
Authors reply: With the help of this experimental set-up, it could be shown that there is obviously a connection between the activation of Piezo1 and TRPV4. Since it is postulated in the literature that these two channels are involved in the sensitivity to mechanical stress and it is discussed that there might be an imbalance here under OA, this experiment was aimed at characterizing initial physiological connections, i.e. connections on a cellular level.
Fig7 shows the analysis of the gene expression. What are the protein expression, especially for the signalling molecules, as well as their phosphorylation status?
Authors reply: Thank you very much for your valuable comments! These are preliminary experiments to find indications of changes in different signaling pathways/directions to serve as a basis for further investigations, especially for planned experiments regarding protein phosphorylation.
Overall, this manuscript needs to be subjected to major revision through additional experiments and data presentation and re-written.
We thank Reviewer 2 for her/his time and efforts.
Round 2
Reviewer 2 Report
Thanks to the authors for revising the manuscript which is improved to some extent. However, there are still some major concerns regarding the current version of the manuscript.
Data analysis and presentation:
The authors used ‘n’ in the figure legend to express both the technical replicates and donors which is confusing. Does this mean that those dots or figures in parentheses, e.g. shown in Fig1d, represent the mixture of both? In theory, 4 dots of biological replicates should be displayed in each bar in this graph here and each dot represented the average of technical measures) if this study analyzed 4 donors (Pg3, line 133-134). If the authors analyzed 'truly' n=8 replicates and n=4 donors, the number of replicates in total should be 32 (line 136), and apparently, this isn't the case as different replicates were shown in the figure. So, unfortunately, the description in the figure legend still does not truly reflect the actual data.
The number of experimental replicates for different settings, i.e. qPCR and IF, should be described separately in the figure legend, rather than mixed as n=2 and 4.
Can Fig4 c&d be moved to Fig1 because they do not fit in Fig4 about ‘Expression of mechanosensitive channels in pCH-OA and HC’?
Style of the legend description:
Should the authors follow the standard style/order, such as a), b), c), etc. followed by the description after the title of the figures? It is difficult to follow the descriptions in the current version.
e.g. Fig3: (d) is missing from the figure legend
Other issues:
Pg1, line 42: please check the figure, is the calculation of 113.25% correct?
Pg3, line 121: what does HC stand for? The full text should be shown in the first place along with the abbreviation. If the authors have done the actual experiments with HC cells, the data should be shown, at least in supplementary info, rather than just adding the sentence as these data from HC cells is important even though they looked comparable with the OA cells.
What expression changes did the authors measure here in ‘2.5. Changes in expression by Yoda1 in pCH-OA and HC’?
Author Response
Thanks to the authors for revising the manuscript which is improved to some extent. However, there are still some major concerns regarding the current version of the manuscript.
Data analysis and presentation:
- The authors used ‘n’ in the figure legend to express both the technical replicates and donors which is confusing. Does this mean that those dots or figures in parentheses, e.g. shown in Fig1d, represent the mixture of both? In theory, 4 dots of biological replicates should be displayed in each bar in this graph here and each dot represented the average of technical measures) if this study analyzed 4 donors (Pg3, line 133-134). If the authors analyzed 'truly' n=8 replicates and n=4 donors, the number of replicates in total should be 32 (line 136), and apparently, this isn't the case as different replicates were shown in the figure. So, unfortunately, the description in the figure legend still does not truly reflect the actual data
Authors respond: We apologize for the confusion and unclear presentation, but we interpreted the reviewer’s remark in the first revision as to indicate both the number of donors and replicates in the figures and figure legend. Accordingly, the numbers given in the figures and also in parentheses represent the total number of experiments performed.
We have now specified this more precisely in the figure legends in the following way:
- For Figure 1(d) we removed the following sentence from the figure legend: “Numbers of replicates are given in parenthesis; number of donors n = 4. Different treatment options did not change the response of pCH-OA to the different Yoda1 concentrations (e); replicates n = 8, donors n = 4.
Therefore we added the following comment: „The corresponding experiments were performed with four different donor cell preparations (preparations of primary chondrocytes from four different OA patients). For each donor cell preparation, the individual treatment conditions were experimentally repeated with different frequencies; this number of experiments is indicated in brackets in the figure in each case“(Line 139-142).
- For Figure 2 we removed the comment: number of pCH-OA donors throughout the whole experiment n = 8.
Instead we added: For c), d), e) the total amounts of repeated measurements on at least four different donor cells are given in parenthesis.
Note to reviewer: the number of replicates was not the same for all donors, hence the difference in (n) in Fig 2.e; but as we stated in the figure legend the experiments were NOT performed on only one donor except for HC.
- Figure 3/4: To counteract the confusion, we added the following text to the figure legend: „The pCH-OA cells used came from eleven (in Fig.3)/eight (in Fig.4) different donors and were used with varying frequency throughout this set of experiments“ plus added the term „total“ to experiments (Line 210-212; Line 251-256)
- Figure 5/6: Lines 301-303, 322-323, and 362-371 in the figure legends have been adjusted accordingly in the same comparative manner.
- The number of experimental replicates for different settings, i.e. qPCR and IF, should be described separately in the figure legend, rather than mixed as n=2 and 4.
Authors respond: Again, we would like to apologize for this oversight and this ambiguity has been adjusted – see Line 245–246.
- Can Fig4 c&d be moved to Fig1 because they do not fit in Fig4 about ‘Expression of mechanosensitive channels in pCH-OA and HC’?
Authors respond: If possible, we would prefer to leave these two plots in Figure 4, as this would complete the picture of the expression of these two important mechanosensitive ion channels. This would also allow a good comparison of the intrinsic activity of Piezo1 between healthy and diseased cells and show that there is neither a massive difference in the expression nor in the chemical activability of this particular channel, in contrast to TRPV4.
- Style of the legend description:
Should the authors follow the standard style/order, such as a), b), c), etc. followed by the description after the title of the figures? It is difficult to follow the descriptions in the current version. e.g. Fig3: (d) is missing from the figure legend
Authors respond: We are sorry if there have been any difficulties in understanding here. In most cases, the order has been followed, otherwise, as noted by the reviewer's valuable comment, it may be difficult to follow the descriptions of the individual figures. In exceptional cases, there may have been shifts due to better understanding and in the description of the results.
For Fig3 (d): We apologize for this omission and error and have added the correct designation.
- Other issues:
- Pg1, line 42: please check the figure, is the calculation of 113.25% correct?
Authors respond: Thank you for this thoughtful correction to the work. The number increased from 247 million to 527 million, a difference of 280 million, and if you put this in relation to the baseline in the year 1990 (i.e., 247 million), it represents a 113% increase. For a better understanding, we have changed this description to: "... has more than doubled".
- Pg3, line 121: what does HC stand for? The full text should be shown in the first place along with the abbreviation. If the authors have done the actual experiments with HC cells, the data should be shown, at least in supplementary info, rather than just adding the sentence as these data from HC cells is important even though they looked comparable with the OA cells.
Authors respond: HC is the abbreviation for “healthy chondrocytes”; the abbreviation was explained at the first mention and in the methods section under 4.1 Cartilage samples and cell culture (line 460).
- What expression changes did the authors measure here in ‘2.5. Changes in expression by Yoda1 in pCH-OA and HC’?
Authors respond: Many apologies for this unclear headline. We have changed the subtitle: “Yoda 1 induced changes in the expression of MMPs, TIMP-2, BMP-2, Col1A1, and Interleukins”.
Additional activities by the authors:
- We have additionally written out the abbreviation OA in the title
- We have completed the naming of the authors and added R. Windhager
- In the Abstract we changed the following sentence: “Therefore, it is important to consider that the extent of mechanical stimulation may have different effects on OA” to “Therefore, it is important to consider that mechanical stimulation may have different effects in OA depending on its intensity”.
Round 3
Reviewer 2 Report
Re. Pg1, line 42: "...increased by 113.25% ...". Thanks to the authors for the clarification. So it is up to you to keep the original sentence or not.